# The GAG-Binding Peptide MIG30 Protects against Liver Ischemia-Reperfusion in Mice

**DOI:** 10.3390/ijms23179715

**Published:** 2022-08-26

**Authors:** Thiago Henrique Caldeira Oliveira, Vincent Vanheule, Sofie Vandendriessche, Fariba Poosti, Mauro Martins Teixeira, Paul Proost, Mieke Gouwy, Pedro Elias Marques

**Affiliations:** 1Immunopharmacology Laboratory, Department of Biochemistry and Immunology, Institute of Biological Sciences, Universidade Federal de Minas Gerais, Belo Horizonte 31270-901, Brazil; 2Laboratory of Molecular Immunology, Department of Microbiology, Rega Institute, Immunology and Transplantation, KU Leuven, 3000 Leuven, Belgium

**Keywords:** liver, ischemia-reperfusion, chemokines, glycosaminoglycans, neutrophils, inflammation, CXCL9

## Abstract

Ischemia-reperfusion injury (IRI) drives graft rejection and is the main cause of mortality after liver transplantation. During IRI, an intense inflammatory response marked by chemokine production and neutrophil recruitment occurs. However, few strategies are available to restrain this excessive response. Here, we aimed to interfere with chemokine function during IRI in order to disrupt neutrophil recruitment to the injured liver. For this, we utilized a potent glycosaminoglycan (GAG)-binding peptide containing the 30 C-terminal amino acids of CXCL9 (MIG30) that is able to inhibit the binding of chemokines to GAGs in vitro. We observed that mice subjected to IRI and treated with MIG30 presented significantly lower liver injury and dysfunction as compared to vehicle-treated mice. Moreover, the levels of chemokines CXCL1, CXCL2 and CXCL6 and of proinflammatory cytokines TNF-α and IL-6 were significantly reduced in MIG30-treated mice. These events were associated with a marked inhibition of neutrophil recruitment to the liver during IRI. Lastly, we observed that MIG30 is unable to affect leukocytes directly nor to alter the stimulation by either CXCL8 or lipopolysaccharide (LPS), suggesting that its protective properties derive from its ability to inhibit chemokine activity in vivo. We conclude that MIG30 holds promise as a strategy to treat liver IRI and inflammation.

## 1. Introduction

The interruption of blood flow (ischemia) with consequent lack of oxygen and nutrient supply is an inherent phenomenon during various surgical procedures [1]. In liver surgery, there are particularly long periods of ischemia such as during removal of liver tumors, in trauma, vascular reconstruction and transplantation [2,3,4]. Once the blood flow, oxygen tension and nutrients are restored (reperfusion), there is an increase in injury initiated by ischemia, aggravating the overall damage [5,6,7,8]. This phenomenon is known as ischemia-reperfusion injury (IRI), a biphasic disorder in which the ischemic damage is amplified during the reperfusion phase. Indeed, IRI affects liver viability and directly correlates to graft rejection, a major clinical limitation in orthotopic liver transplantation [9].

Although there is tissue damage during the period of ischemia, which is mostly caused by metabolic stress and reactive oxygen species production, the majority of liver injury occurs during reperfusion, when there is an intense activation of the immune system. In this regard, liver injury induced by IR is linked to intense leukocyte infiltration, in which neutrophils, the first cells to arrive at the site of injury, amplify tissue damage by producing pro-inflammatory mediators, proteases and free radicals [10]. The multi-step process for neutrophils to reach the liver parenchyma and promote additional damage includes neutrophil activation, adhesion within the hepatic vasculature and release of granular content [11,12]. The activation of the immune system is initially induced by resident cells that recognize endogenous molecules called danger-associated molecular patterns (DAMPs) released by necrotic cells. Kupffer cells recognize DAMPs and are stimulated to produce inflammatory cytokines and chemokines, which drive the activation and migration of neutrophils from blood to the area of tissue injury [13,14]. In the injured liver, neutrophils also recognize DAMPs using receptors such as TLR9, which binds to DNA, an abundant DAMP released from dead hepatocytes [15]. These events lead to the upregulation of adhesion molecules such as CD11b/CD18, which promote the adhesion and migration of neutrophils in the liver during IRI.

Among the mediators produced in liver IRI, CXC chemokines play a key role in guiding neutrophils to the site of injury. They form a gradient along which neutrophils become activated and migrate from blood vessels to the site of inflammation. To create this gradient and to induce neutrophil migration in vivo, it is necessary for chemokines to interact with two essential molecules/receptors. First, they need to bind to cell surface glycosaminoglycans (GAGs) at the vascular endothelium and in the extracellular matrix. This interaction is important for preventing chemokine diffusion and retaining high chemokine concentrations locally. Subsequently, GAG-bound chemokines interact using their N-terminal portion with G protein-coupled receptors (GPCRs), which are expressed by multiple hepatic cells, including leukocytes [16,17]. Chemokine–GPCR interaction induces leukocyte adhesion to the endothelium, followed by their directional migration towards inflammatory sites [18]. In this context, pharmacological blocking of the chemokine receptors CXCR1 and CXCR2 with reparexin, an allosteric receptor antagonist, was able to reduce neutrophil infiltration into the liver, which was associated with better liver function and histological outcome [19].

Recently, our group developed a peptide consisting of the COOH-terminal portion of the chemokine CXCL9, which was characterized as a potent GAG-binding peptide [20,21]. The CXCL9(74-103) peptide, also called MIG30, inhibits the chemotactic activity of chemokines by competing with the binding to GAGs. The blockage of GAG–chemokine interactions by MIG30 reduced tissue injury and inflammation in mouse models of drug-induced liver injury [22], monosodium urate-induced gout [21] and bacterial pneumonia [23], among others. These findings illustrate the importance of GAGs in promoting the function of chemokines. However, this phenomenon has been poorly explored in vivo, especially in organs with atypical vasculature such as the liver. Although therapeutic intervention in the chemokine system has long been focused on the development of GPCR antagonists [24], currently, it has been suggested that interference of GAG–chemokine interactions might represent an alternative and innovative way to block chemokine action and decrease inflammation more effectively [25]. Therefore, this work aimed to evaluate the therapeutic potential of a GAG-binding peptide on the severity of liver IRI.

## 2. Results

### 2.1. MIG30 Inhibits the Binding of CXCL6 to Heparan Sulfate GAGs

In addition to receptor binding, chemokine activity largely depends on interactions with GAGs such as heparan sulfate (HS) [26]. Previous work from our group showed that the MIG30 peptide (CXCL9(74-103)) is a powerful competitor of chemokine binding to GAGs, inhibiting binding of CXCL1, CXCL8, CXCL11 and CCL2 to HS [21]. Here, we tested the ability of MIG30 to inhibit binding of CXCL6, the most powerful neutrophil chemoattractant chemokine [27,28], to HS using GAG binding plates in an ELISA-like assay (Figure 1). More than half of the 30 amino acids in MIG30 are positively charged (mainly Lys), and the peptide contains multiple GAG-binding motifs, making it avid to negatively charged GAGs as HS. As expected, MIG30 competed strongly with CXCL6 for binding to HS. Interestingly, even at the lowest molar ratio (1:2), the peptide inhibited more than 60% of CXCL6 binding to HS. When the molar concentration of MIG30 exceeded the concentration of CXCL6 by a factor more than 10, CXCL6 binding was almost completely abolished (Figure 1). In summary, MIG30 inhibits the binding of multiple CXC chemokines to HS, including CXCL6, which may contribute to its anti-inflammatory role in vivo.

### 2.2. Treatment with MIG30 Protects Mice from Liver Ischemia-Reperfusion Injury (IRI)

Considering the clinical importance of liver IRI worldwide and the literature supporting a role for chemokines in IRI severity [14], we were eager to test if treatment with MIG30 would reduce liver injury in the mouse model, in which the hepatic blood supply was partially blocked for 1 h (ischemia) and restored for a period of 6–12 h (reperfusion). Initially, we determined the extent of liver damage induced by our hepatic IRI model using confocal intravital microscopy. We observed that IRI livers presented a broad and massive accumulation of extracellular DNA (Figure 2A). Large blobs of DNA were found scattered in between Sytox green-positive nuclei all around the liver, which are indicative of substantial injury. Although similar in extent, it is interesting to note that these extracellular DNA deposits differed in morphology from the extracellular DNA found during drug-induced liver injury [22]. Quantification of alanine aminotransferase (ALT) levels in the sera of mice confirmed significant liver damage 12 h into the reperfusion phase (Figure 2B). With evidence that the IR model was inducing severe liver injury, we proceeded to treat mice with 100 µg of MIG30 peptide per dose, intravenously, 15 min before the reperfusion and again 6 h later. Treatment with MIG30 abrogated liver injury, as indicated by the significant decrease in ALT levels at 12 h of reperfusion (Figure 2B). The peptide also rescued liver function during IR in mice, as showed by improved clearance of indocyanine green (ICG) from the bloodstream of MIG30-treated mice when compared to saline-treated mice (Figure 2C). Moreover, histopathology analysis (H&E) showed that the general hepatic damage induced by IR, including the sinusoidal congestion and areas of coagulative necrosis, was largely inhibited by treatment with MIG30 peptide (Figure 2D,E). Altogether, these data show that MIG30 reduces significantly the liver injury and dysfunction induced by IR in mice.

### 2.3. MIG30 Treatment Impairs Neutrophil Recruitment during Liver IRI

Neutrophil infiltration is a hallmark of liver IRI [14]. Neutrophil recruitment to the liver occurs during the reperfusion phase and requires the chemokine receptors CXCR1 and CXCR2, among other factors [19,29]. As expected, liver IRI induces neutrophil recruitment to the liver at both 6 and 12 h of reperfusion, as shown by the increased myeloperoxidase (MPO) activity in the hepatic tissue (Figure 3A), which serves as an indirect indicator of the accumulation of neutrophils. These results were confirmed by the increased number of neutrophils (Ly6G^+^ cells) counted in liver cryosections of IRI mice (Figure 3B,C). Interestingly, neutrophil recruitment to the liver was significantly inhibited by MIG30 12 h post reperfusion, as demonstrated by the reduced hepatic MPO activity and decreased number of Ly6G^+^ neutrophils in liver cryosections (Figure 3A–C). In summary, MIG30 is able to inhibit neutrophil recruitment to the liver during IRI.

### 2.4. The Levels of CXC Chemokines and Proinflammatory Cytokines Are Reduced in MIG30-Treated Mice

It has been long appreciated that ELR^+^ CXC chemokines promote neutrophil activation and chemotaxis [14,18]. We hypothesized that the reduction in liver injury and neutrophil recruitment in MIG30-treated mice was a consequence of the interference of MIG30 on the chemokine activity in vivo. Thus, we assessed the levels of three major murine neutrophil activating chemokines: CXCL1, CXCL2 and CXCL6. We observed that all chemokines were significantly increased in the serum of mice subjected to liver IRI 6 h after reperfusion (Figure 4A–C). Treatment with MIG30 reduced significantly the concentration of all three chemokines in murine sera. In line with that, we investigated the levels of two proinflammatory cytokines involved in the acute response triggered by liver IR, i.e., IL-6 and TNF-α, in order to evaluate whether MIG30 could also affect other key mediators. We observed that liver IR induced the production of IL-6 at 6 h post-reperfusion and of TNF-α 12 h post-reperfusion. Interestingly, the treatment with MIG30 was able to prevent the increase of both IL-6 and TNF-α levels in the sera of mice (Figure 4D,E). These data suggest that the protective and anti-inflammatory properties of MIG30 likely derive from its ability to interfere with CXC chemokine and cytokine levels in mice subjected to IRI.

### 2.5. The MIG30 Peptide Does Not Affect the Responsiveness or Activation Status of Leukocytes towards Stimuli

The inhibition of neutrophil recruitment and the reduction in chemokines and cytokines in mice treated with MIG30 led us to hypothesize that the peptide could be affecting leukocytes directly, besides its effects on chemokine and cytokine levels. To test this, we stimulated purified human neutrophils and PBMCs with combinations of CXCL8, LPS and the MIG30 peptide and assessed if the peptide had direct effects or interfered with leukocyte activation. To evaluate potential effects on GPCR signaling, we measured the increase in intracellular calcium in primary human neutrophils stimulated with CXCL8. Stimulation with CXCL8 induces a clear elevation in intracellular calcium, which does not occur in neutrophils stimulated with buffer or MIG30 alone (Figure 5A). Pre-incubation of neutrophils with MIG30 peptide did not alter the intensity of the calcium signal induced by CXCL8. Similarly, stimulation of neutrophils with a combination of both CXCL8 and MIG30 resulted in the same calcium signal as observed on cells induced by CXCL8 alone (Figure 5A and Appendix A). These data show that MIG30 does not induce an increase in intracellular calcium directly, nor does it affect the calcium signal induced by chemokine receptor activation by CXCL8 in neutrophils. Next, we evaluated if the production of CXCL8 by human PBMCs could be altered in the presence of MIG30. PBMCs stimulated with buffer or MIG30 secrete a similarly low level of CXCL8 in the supernatant. However, upon stimulation with LPS, the level of CXCL8 in the supernatant was substantially increased (Figure 5B). Co-incubation of PBMCs with LPS and MIG30 was unable to alter significantly the production of CXCL8. These data suggest again that MIG30 does not stimulate leukocytes directly and that it does not inhibit the activity of other established pro-inflammatory stimuli.

We also evaluated if the expression of surface receptors in neutrophils was altered by MIG30. Using flow cytometry, we compared the activation status of neutrophils stimulated by LPS, CXCL8 or MIG30. In general lines, for all the receptors evaluated (CD62L, CD11b, CXCR2, CD35, CD16 and CD66b), we did not observe any difference in expression when comparing buffer and MIG30-stimulated neutrophils (Figure 5C). This lack of stimulatory effect by MIG30 is made even clearer when comparing to LPS-treated neutrophils, used as a positive control, in which all the above-mentioned markers were significantly altered. Corroborating our data in Figure 5A, we also observed that co-incubation of cells with CXCL8 and MIG30 was essentially equal to stimulation with CXCL8 alone for all receptors. In more detail, stimulation of neutrophils with CXCL8 or LPS promotes the shedding of the selectin CD62L, a sign of neutrophil activation. Co-incubation with CXCL8 and MIG30 had a very similar effect to CXCL8 alone, whereas incubation of neutrophils with MIG30 only did not cause any CD62L downregulation. CD11b, another adhesion molecule, was significantly upregulated by LPS, and only a trend in increased expression of CD11b was observed in neutrophils stimulated with CXCL8 and CXCL8+MIG30. CXCR2 expression followed the same pattern as CD62L; it was downregulated (internalized) after stimulation with CXCL8, CXCL8+MIG30 and LPS, indicating similar degrees of neutrophil activation by each and all stimuli. CD35, or complement receptor 1, was marginally upregulated by CXCL8 stimulation (with or without MIG30), but significantly upregulated by LPS. CD16, an Fcγ receptor, was unaltered by all stimuli except for its downregulation by LPS. CD66b, a membrane protein from secondary granules, was modestly increased by CXCL8 and CXCL8+MIG30 stimulation but significantly upregulated by LPS. Altogether, these analyses show that MIG30 is unable to activate neutrophils and that it does not alter the stimulation by CXCL8.

Lastly, we investigated if the release of granular content by neutrophils could be induced or affected by MIG30. Corroborating the lack of effect on calcium signaling, CXCL8 production and receptor expression (Figure 5A–C), MIG30 does not induce a significant release of elastase nor MPO from neutrophils when incubated alone or in combination with CXCL8 (Figure 5D,E). LPS was also used as positive control in these experiments, and it induces a significant increase in the concentration of both elastase and MPO when compared to buffer-treated cells. In summary, the data in Figure 5 show that MIG30 is not affecting neutrophil activation or responsiveness to inflammatory stimuli, such as the chemokine CXCL8. It suggests that the anti-inflammatory and protective effects by MIG30 derive mostly from its interference on chemokine activity in vivo, not from cell-intrinsic inhibitory effects.

## 3. Discussion

In this study, we investigated the protective and anti-inflammatory effects of MIG30, a potent GAG-binding peptide. Treatment of mice subjected to liver IRI with MIG30 reduced significantly liver injury, hepatic dysfunction and inflammation. These protective effects were associated to a significant decrease in chemokine levels and neutrophil recruitment to the liver, which contribute to IRI severity [14,19,29]. The participation of neutrophils in IRI involves multiple events such as cellular activation by DAMPs released from necrotic cells, chemokine secretion, chemokine binding to GAGs, expression of adhesion molecules by endothelial cells and leukocytes, adhesion and migration of neutrophils and release of effector molecules. The manipulation of these steps may expand our understanding of the immune response to injury and may open venues to improve both the recovery from IRI and overall graft success in transplantation.

The binding of chemokines to GAGs has been proven indispensable for chemokine activity. Its importance was initially supported by studies demonstrating binding of chemokines to purified GAGs in vitro and in vivo [30,31]. Substantial evidence was further shown in mice deficient in HS sulfation [32] and in wild-type mice injected with mutant chemokines unable to bind GAGs [33]. In the former, lack of functional HS rendered endothelial cells incapable of immobilizing chemokines properly and caused a significant defect in neutrophil migration to air pouches in response to CXCL1, CXCL2 or LPS [32]. In the latter, chemokines mutated at known GAG-binding sites fail to recruit leukocytes to the mouse peritoneum, although the chemokines still largely retain their chemotactic activity in vitro, demonstrating how critical the GAG-chemokine interaction is to leukocyte migration in vivo [33]. The MIG30 peptide follows on the same principle; it competes with chemokine binding to HS in vitro (Figure 1), probably disrupting its capacity to form gradients that drive neutrophil activation and recruitment in vivo (Figure 3 and Figure 4). Binding of chemokines to GAGs can also be inhibited by other agents, such as intact modified chemokines [25]. These modified chemokines have an enhanced affinity for GAGs in comparison with their natural counterparts but a decreased affinity for the GPCRs. In this way, the modified chemokines compete with functional chemokines for GAG binding, immobilization and presentation, inhibiting chemokine-induced leukocyte migration. In line with that, PA401, a CXCL8-based decoy protein, exerts strong anti-inflammatory activity in vivo [34,35]. These molecules illustrate that interfering with the chemokine–GAG interactions might constitute an attractive way of treating chemokine-mediated disorders and act as a complementary option for chemokine receptor antagonism.

It is worth noting that the protective effect of MIG30 in the murine IRI model (Figure 2) was more pronounced than in the drug-induced liver injury model [22]. We believe that the window of treatment and duration of the model play an important role in this difference. In the IRI model, the experiment lasted for a maximum of 12 h and the mice were treated immediately before the reperfusion phase and once again at 6 h. On the other hand, in the drug-induced liver injury model, the mice were treated 6 h after the toxic insult and treated again 2 more times for the following 18 h. It is plausible that the immediate bioavailability of MIG30 at the moment of injury, added to the shorter experimental time, maximized its therapeutic efficacy when compared to the drug-induced liver injury model. Moreover, this implies that MIG30 clearance may significantly impact its effects in longer/chronic diseases. Modifications to the peptide sequence to inhibit its clearance by the kidneys (e.g., PEGylation) and to reduce proteolytic degradation (e.g., N-terminal acetylation and C-terminal amidation) might be useful to extend its therapeutic effects. These remarks should be considered when testing or developing other GAG-binding agents in mouse models.

We observed that mice treated with MIG30 presented significant reductions in the levels of CXC chemokines and of cytokines (Figure 4). The reduction in the serum levels of CXCL1, CXCL2 and CXCL6 illustrates one of the advantages of using a GAG-binding reagent: it might inhibit the binding of multiple chemokines simultaneously. We also speculate that the lack of anchoring of chemokines to GAGs, as induced by MIG30 in the liver, predisposes them to proteolytic degradation [36] and clearance by atypical chemokine receptors [37], causing the chemokine levels to be more rapidly reduced in MIG30-treated mice. Lastly, it is reasonable to assume that the early impairment in chemokine activity and neutrophil recruitment prevented the injury worsening during the reperfusion phase, as illustrated by reduced TNF-α levels at 12 h (Figure 4E) and the overall better outcome of MIG30-treated compared to saline-treated mice (Figure 2). All these point to the notion that controlling chemokine and neutrophil activity early in the reperfusion phase may have a broader positive impact than anticipated.

An important point to consider with any treatment is its unspecific (or unexpected) effects. In Figure 5, we performed an extensive evaluation of potential leukocyte activation in response to MIG30 in the presence or absence of other stimuli. It was clear that the peptide does not affect CXCL8 signaling in neutrophils, that it neither induces or inhibits the production of CXCL8, and that it does not upregulate or downregulate neutrophil activation markers. In other words, it does not affect neutrophil function directly, pointing to a main dominant effect on the chemokine-GAG interaction in the liver in vivo. Other evaluations about off-target effects of MIG30 were performed previously in neutrophils, PBMCs and endothelial cells, such as its capacity to induce neutrophil extracellular traps or to affect reactive oxygen species levels (ROS) [22]. Moreover, in these conditions, MIG30 was incapable to cause any significant alteration. These observations suggest that MIG30 is generally inert to cells, which is somewhat surprising for a positively charged peptide but still in agreement with the homeostatic and immunological roles of CXCL9. An interesting future step would be to study if other C-terminal portions of chemokines, especially those positively charged, share similar properties with MIG30 in vivo.

## 4. Materials and Methods

### 4.1. MIG30 Peptide Synthesis

The GAG-binding CXCL9 fragment (CXCL9(74-103); MIG30) was chemically synthesized using Fmoc (N-(9-fluorenyl)methoxycarbonyl) chemistry on an Activo-P11 automated solid phase peptide synthesizer (Activotec, Cambridge, UK) [20,21]. After synthesis, intact synthetic MIG30 was purified by high-performance liquid chromatography (HPLC) and characterized by electrospray mass spectrometry using an ion trap Amazon SL instrument (Bruker). Purified peptides were lyophilized resuspended in sterile PBS and stored at −80 °C until use.

### 4.2. Competition Assay for Binding of CXCL6 to GAGs In Vitro

The ability of MIG30 to compete for GAG binding with the chemokine CXCL6 was evaluated on heparin-binding plates ((BD Biosciences, Erembodegem, Belgium) or kindly provided by Dr. Jason Whittle (University of South Australia, Australia)) [38]. In brief, wells were coated with heparan sulfate (HS) (25 µg/mL) overnight at room temperature. Dilutions of recombinant murine CXCL6(9-72) (300, 100, 30, 10 and 3 nM; Peprotech Rocky Hill, NJ, USA) were added to heparan sulfate plates in the presence or absence of MIG30 (100 nM). Samples were added in duplicate and incubated for 2 h at 37 °C. Subsequently, bound CXCL6 was detected with biotinylated polyclonal goat anti-mouse CXCL6 (Peprotech, Rocky Hill, NJ, USA) and peroxidase-labeled streptavidin. The peroxidase activity was quantified using 3,3′-5,5′-tetramethylbenzidine (TMB) substrate conversion and the absorbance was read at 450 nm.

### 4.3. Model of Hepatic Ischemia-Reperfusion Injury (IRI)

C57BL/6J male mice (8–12 weeks old) were provided by the Animal Facility of the Rega Institute (KU Leuven). The animals were maintained with filtered water and food ad libitum in a 12 h dark–light cycle in the thermoneutral zone for mice. All experiments were approved by the ethical committee for animal experiments from KU Leuven (P111/2016). IR was performed as described [19]. C57BL/6J mice were anesthetized with an intraperitoneal injection of ketamine (80 mg/kg) and xylazine (4 mg/kg). After a midline laparotomy, mice underwent a sham control operation or IR. In the IR group, the pedicle of the left and median lobes of the liver, containing the bile duct, hepatic artery, and portal vein (comprising 70% of the liver) was occluded using an atraumatic clamp (Aleamed, Kontich, Belgium). After 60 min of ischemia, the clamp was removed, and reperfusion was initiated. The time points of 6 h and 12 h were examined after reperfusion. The control operation was performed with the same protocol but without vascular occlusion. In this case, the sham group refers to animals operated in the earliest time-point evaluated in each experiment (6 h), since we observed no difference between sham groups at any timepoint after surgery, in any of the parameters evaluated (data not shown). Mice were placed on a heating pad to maintain body temperature at 37 °C throughout the procedure. Mice were treated with 100 µg of MIG30 peptide per dose, intravenously, 15 min before the reperfusion and again 6 h later. Liver injury was estimated by serum ALT activity using a kinetic assay by Teco Diagnostics. Cytokines (i.e., IL-6, TNF-α) and chemokines (CXCL1, CXCL2, CXCL6) were quantified by enzyme-linked immunosorbent assay kits (R&D Systems, Minneapolis, MN, USA) in murine sera. Livers were collected for quantitative polymerase chain reaction (qPCR) analysis of cytokine expression. Fragments of liver were fixed and sectioned for histology, as described below. Indocyanine green (ICG; Sigma-Aldrich, St. Louis, MA, USA) clearance by the liver was estimated in serum after injecting a single dose of 20 mg/kg intravenously. Blood was collected 20 min after injection, and the amount of ICG was determined by spectrophotometry (absorbance in 800 nm).

### 4.4. Liver Intravital Microscopy

Confocal liver intravital microscopy was performed as described previously [39]. Livers were imaged by intravital microscopy using a Nikon Eclipse Ti microscope (Langen, Germany) with a C2 confocal head and equipped with a 10× objective. Mice were anesthetized with ketamine/xylazine, and the liver was surgically exposed on an acrylic stage compatible with the microscope. Sytox green was used to stain extracellular DNA/injury sites (2 µL/mouse at 5 mM; Invitrogen, Waltham, MA, USA) and was injected intravenously 10 min before imaging.

### 4.5. Histological Analysis

The livers were washed with 0.9% NaCl and fixed in 4% buffered formalin. Subsequently, the samples were dehydrated in ethanol, bathed in xylol and embedded in histological paraffin blocks. Tissue sections of 5μm thickness were obtained using a microtome and stained with hematoxylin & eosin (H&E). The slices were visualized using the BX41 (Olympus, Center Valley, PA, USA) optical microscope, and images were obtained using the Moticam 2500 camera (Motic, Barcelona, Spain) and Motic Image Plus 2.0ML software (Motic, Barcelona, Spain). H&E-stained sections were observed for any abnormalities of histopathological features. The degree of hepatic injury was based on the grading system as previously described [40]: 0 (Normal: no hepatocytes necrosis); 1 (Minimal: mild focal, limited to centrilobular region, less than ¼ of affected lobules are necrotic); 2 (Mild-moderate: focal and multifocal central to midzonal lobular region, ½ affected lobules are necrotic); 3 (Severe: multifocal, more than ¾ of affected lobules are necrotic).

### 4.6. Myeloperoxidase (MPO) Activity Assay

The extent of neutrophil accumulation in the liver was estimated by tissue MPO activity as performed previously [41]. The tissue was collected and frozen at −70 °C. Upon thawing, 0.1 g tissue per 1.9 mL buffer (0.1 M NaCl, 0.02 M Na_2_PO_4_, 0.015 M Na_2_EDTA, pH 4.7) was homogenized, centrifuged at 3000× *g* for 10 min, and subjected to hypotonic lysis with 0.2% NaCl for 30 s followed by an equal volume of 1.6% NaCl with 5% glucose. After centrifugation, the pellet was resuspended in 0.05 M Na_2_PO4 (pH 5.4) containing 0.5% hexadecyl-trimethylammonium bromide (HTAB) and re-homogenized. After a 15 min centrifugation at 3000× *g*, the supernatant was harvested and used in the assay. The MPO assay combines 25 µL of TMB substrate (Sigma-Aldrich) dissolved in dimethyl sulfoxide at a final concentration of 1.6 mM, 100 µL of 0.002% H_2_O_2_ in PBS pH 5.4 containing HTAB and 25 µL of processed sample supernatant. The reaction was performed at 37 °C for 5 min in a 96-well microplate by adding the supernatant and the TMB solution. After that, H_2_O_2_ was added and followed by a new incubation at 37 °C for 5 min. The reaction was stopped by adding 100 mL of 1 M H_2_SO_4_ and quantified at 450 nm in a spectrophotometer (Emax; Molecular Devices, San Jose, CA, USA).

### 4.7. Leukocyte Purification (Neutrophils and PBMCs)

Fresh venous blood samples from healthy volunteers were collected in EDTA-coated tubes (BD Biosciences). The isolation of peripheral blood mononuclear cells (PBMCs) and neutrophils was performed by density gradient centrifugation in Pancoll (PAN Biotech GmbH, Aidenbach, Germany). After 30 min centrifugation (400× *g*, 20 °C), the middle layer containing PBMCs was isolated and washed twice with DPBS (10 min, 177× *g*, 20 °C). The pellet, containing peripheral blood neutrophils, was incubated with 6% Dextran (Sigma-Aldrich; 30 min, 37 °C) in DPBS, and a hypotonic shock (30 s) with ultrapure water was performed to eliminate erythrocytes. Human neutrophils purified by density-gradient centrifugation were used in the Ca^2+^ signaling assay. For phenotypical analysis and protease release, neutrophils were purified via immunomagnetic isolation using the EasySep Direct human neutrophil isolation kit (StemCell Technologies, Cambridge, UK) according to the manufacturer’s instructions [42]. The ethical committee of University Hospital (UZ) Leuven approved experiments involving human neutrophils (project S58418).

### 4.8. Ca^2+^ Mobilization Assay

Purified human neutrophils (1 × 10^7^ cells/mL) were incubated for 30 min at 37 °C in RPMI supplemented with 10% fetal calf serum (FCS), 2.5 µM of Fura-2 acetoxymethylester (Fura-2/AM; Invitrogen) and 0.01% Pluronic-F127 (Sigma-Aldrich). After incubation, the cells were washed in RPMI + 10% FCS, centrifuged for 10 min (4 °C, 177× *g*) and suspended in calcium buffer (HBSS containing 1 mM Ca^2+^, 0.1% FCS, 10 mM HEPES/NaOH, pH 7.4). Cells were kept on ice and used at a final concentration of 1 × 10^6^ cells/mL. Fura-2 fluorescence was measured in a LS50B fluorescence spectrometer (Perkin Elmer, Waltham, MA, USA) fitted with a thermostable, stirred, four-position cuvette holder. Cells (1800 µL) were allowed to equilibrate at 37 °C for 10 min before fura-2 fluorescence intensity was measured. The second stimulus was added 70 s after the first stimulus. To obtain Rmax, cells were subsequently lysed by addition of 50 µM digitonin (Sigma-Aldrich). Finally, Rmin was obtained by the addition of 10 mM EGTA to the lysed cells after adjusting the pH to 8.5 with 20 mM Tris. Intracellular calcium concentrations were calculated using the Grynkiewicz equation [43].

### 4.9. Neutrophil Flow Cytometry

A total of 2 × 10^6^ neutrophils were seeded in a 6-well plate in RPMI medium containing 10% FCS and granulocyte-macrophage colony-stimulating factor (GM-CSF; Peprotech; 5 ng/mL). Cells were stimulated for 1 h at 37 °C with: lipopolysaccharide (LPS from *E. coli*; Sigma-Aldrich, 5 µg/mL), CXCL8(6-77) (Peprotech; 10 ng/mL), MIG30 (20 µg/mL) or a combination of CXCL8 and MIG30. After 1h, the cells were collected and spun down (10 min, 240× *g*) and analyzed for marker/receptor expression. A total of 2 × 10^5^ purified neutrophils were incubated with FcR block (Miltenyi Biotec, Bergisch Gladbach, Germany) and Fixable Viability Stain 620 (BD Biosciences) in PBS for 15 min at RT. Afterwards, cells were washed twice with flow cytometry buffer (PBS + 2% FCS + 2 mM EDTA) and stained with mouse fluorescently labeled antibodies: APC-labeled anti-CD62L (clone DREG56, eBioscience, San Diego, CA, USA), BV510-labeled anti-CD11b (clone ICRF44, BD Biosciences), FITC-labeled anti-CD182/CXCR2 (clone 6C6, BD Biosciences), AF700-labeled anti-CD16 (clone 3G8, BD Biosciences), BV421-labeled anti-CD66b (clone G10F5, BD Biosciences), BV510-labeled anti-CD63 (clone H5C6, BD Biosciences) and FITC-labeled anti-CD35 (clone E11, Biolegend, London, UK). Following incubation for 25 min at 4 °C, cells were washed twice with flow cytometry buffer and fixed with BD Cytofix (BD Biosciences). Samples were run on a BD LSRFortessa equipped with DIVA software. FlowJo software was used for downstream analysis. Neutrophils were gated as CD16^+^CD66b^+^ cells within the population of live, single cells.

### 4.10. Release of CXCL8, Elastase and MPO In Vitro

For the production of CXCL8 in vitro, PBMCs were stimulated for 24 h with 100 ng/mL LPS (Sigma-Aldrich) and/or 4 µg/mL (1 µM) MIG30. The CXCL8 concentration was measured in the supernatants by ELISA. In a 96 well-plate (Corning), 250 ng/mL of the monoclonal anti-human CXCL8 capture antibody (MAB208, R&D systems) was incubated overnight at 4 °C. Plates were blocked with PBS with 0.1% casein and 0.05% Tween 20 for 1 h at 37 °C. Detection was performed with 250 ng/mL biotinylated polyclonal goat anti-human CXCL8 antibody (BAF208, R&D Systems) for 1 h at 37 °C. The incubation of peroxidase-conjugated streptavidin for 30 min at 37 °C allowed the oxidation of TMB substrate in the presence of H_2_O_2_, after which the reaction was stopped by adding 1 M H_2_SO_4_. Absorbance was measured at 450 nm using a PowerWave XS plate reader (BioTek Instruments, Aschaffenburg, Germany) and compared to a standard curve of recombinant CXCL8 (Peprotech). For the release of elastase and MPO, 2 × 10^6^ neutrophils were seeded in a 6-well plate in RPMI-1640 medium containing 10% FCS and GM-CSF (5 ng/mL). Cells were stimulated for 1 h at 37 °C with LPS (5 µg/mL), CXCL8 (10 ng/mL), MIG30 (20 µg/mL) or a combination of CXCL8 and MIG30. After 1 h, the cells were spun down (10 min, 240× *g*) and analyzed for marker/receptor expression (vide supra). Release of elastase and MPO in the supernatants was determined by commercially available Duoset ELISAs (R&D Systems; catalog numbers DY9167 and DY3174), according to the manufacturer’s instructions.

### 4.11. Immunofluorescence Staining

Immunostaining was performed on 4 μm acetone-fixed liver cryosections. To investigate neutrophil infiltration in the inflamed liver, immunofluorescent labeling was performed using PE-coupled rat anti-mouse Ly6G (clone 1A8, BD Bioscience). Hoechst (10 mg/mL) was used for nuclear counterstaining, and sections were mounted with Prolong Gold antifade reagent (Thermo Fisher Scientific). Images were acquired with a Zeiss Axiovert 200M (Carl Zeiss AG, Oberkochen, Germany) and AxioVision Rel 4.8 acquisition software (Carl Zeiss AG).

### 4.12. Statistical Analyses

Experimental data analysis was performed with one-way analysis of variance (ANOVA with Tukey’s post hoc test) and Student *t* test provided by Prism 6.0 software (GraphPad, San Diego, CA, USA). All data are given as the mean ± SEM. In vivo experimental groups had at least four mice per group. Data shown are representative of at least two independent experiments. Differences were considered significant when *p* < 0.05.

## 5. Conclusions

MIG30 is a small, chemokine-derived GAG-binding peptide that has anti-inflammatory and protective effects in murine liver IRI. Intravenous application of MIG30 after the ischemic phase resulted in improved liver function in mice. As such, MIG30 has potential to be used as a treatment to reduce liver tissue damage induced by ischemia and the subsequent reperfusion of the organ.

## Figures and Tables

**Figure 1 ijms-23-09715-f001:**
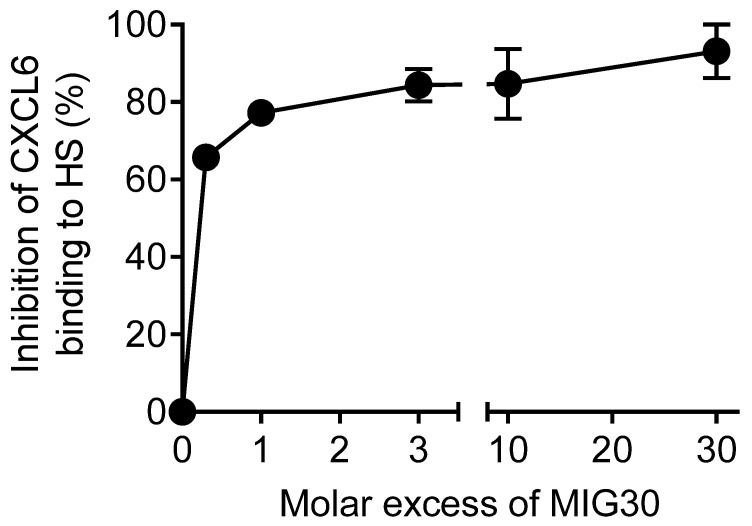
MIG30 inhibits the binding of CXCL6 to heparan sulfate GAGs. Competition between CXCL6 and MIG30 peptide for binding to immobilized heparan sulfate was evaluated on 96-well plates. CXCL6 (300, 100, 30, 10 and 3 nM) was added to heparan sulfate plates in the presence/absence of MIG30 (CXCL9(74-103); 100 nM). Bound CXCL6 was detected with biotinylated anti-mouse CXCL6 antibody. The data shown are the mean ± SEM of the percent of inhibition compared to CXCL6 binding alone.

**Figure 2 ijms-23-09715-f002:**
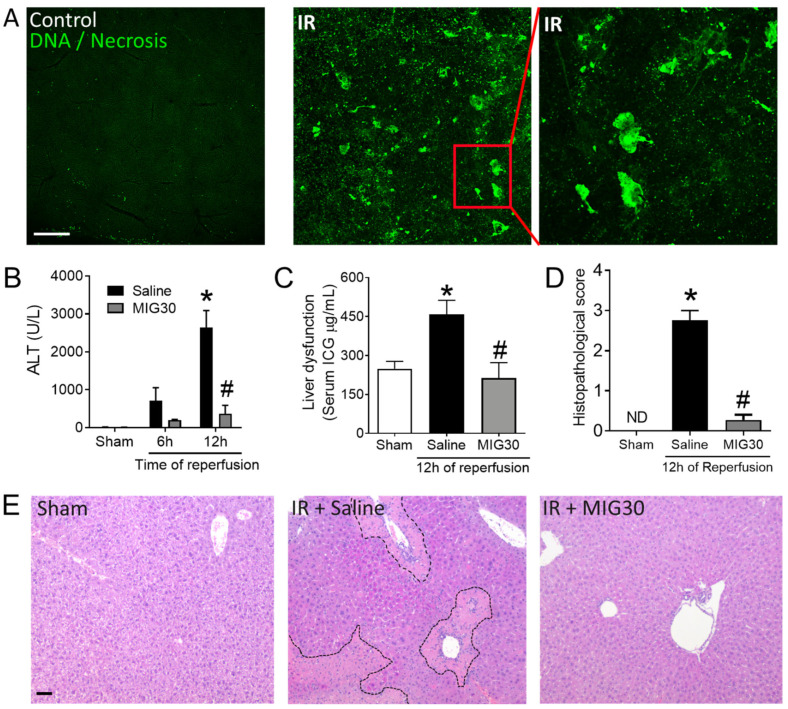
Treatment with MIG30 protects mice from liver ischemia-reperfusion injury (IRI). (**A**) Confocal intravital microscopy of the liver of mice subjected to sham (control) or 8 h of ischemia-reperfusion (IR) injury. Extracellular DNA and injured areas were stained by Sytox green injected intravenously. (**B**) Serum alanine aminotransferase (ALT) levels of mice subjected to different times of IR injury. IR mice were treated i.v. with saline vehicle or with MIG30 (100 µg 15 min before the reperfusion and again 6 h later). (**C**) Serum level of indocyanine green (ICG) showing the severity of liver dysfunction induced by IRI. Mice were treated with saline or MIG30 as in B. (**D**) Histopathology quantification of liver sections stained with H&E. (**E**) Representative H&E staining images of livers 12 h post-IR injury. ND = Not detected. The data shown is mean ± SEM; n is at least 4 mice per group. * *p* < 0.05 vs. sham group; # *p* < 0.05 vs. saline-treated mice. Scale bars = 100 µm.

**Figure 3 ijms-23-09715-f003:**
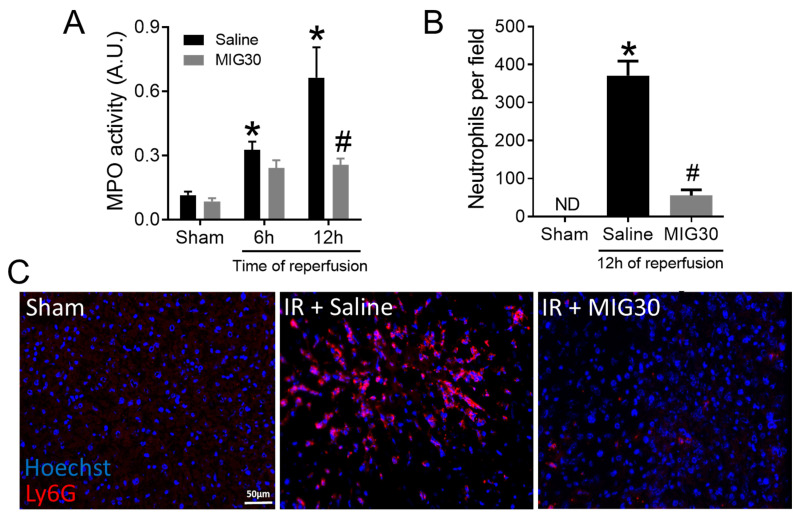
MIG30 treatment impairs neutrophil recruitment during liver IRI. (**A**) Liver MPO activity in mice subjected to liver ischemia-reperfusion injury (IRI). Sham operated mice were used as control. IRI mice were treated i.v. with saline vehicle or with MIG30 (100 µg 15 min before the reperfusion and again 6 h later). (**B**,**C**) Immunostaining of liver cryosections with anti-Ly6G to show recruited neutrophils. Neutrophils (Ly6G^+^) are shown in red and Hoechst-stained nuclei in blue. (**B**) The data shown are the averaged number of neutrophils per field of view. (**C**) Representative photomicrographs of Ly6G^+^ neutrophils in sham, mice subjected to IR and treated with saline or IR-injured mice treated with MIG30. ND = Not detected. The data shown are the mean ± SEM; n is at least 4 mice per group. * *p* < 0.05 vs. sham group; # *p* < 0.05 vs. saline-treated mice. Scale bar = 50 µm.

**Figure 4 ijms-23-09715-f004:**
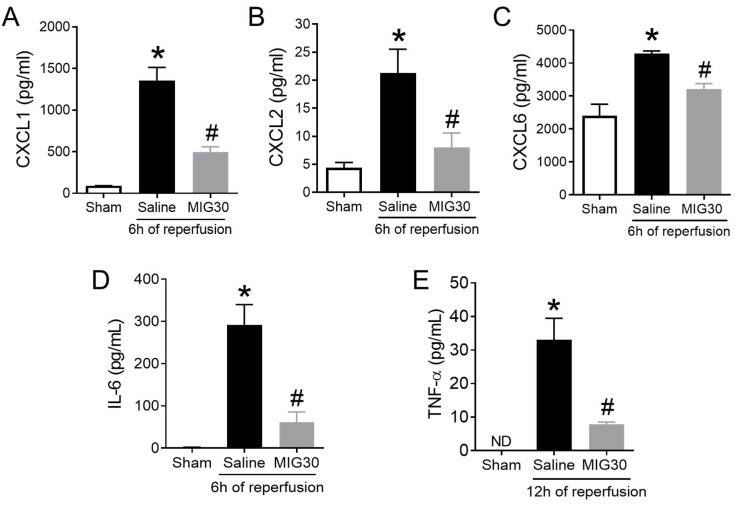
The levels of CXC chemokines and proinflammatory cytokines are reduced in MIG30-treated mice. Quantification of (**A**) CXCL1, (**B**) CXCL2, (**C**) CXCL6, (**D**) IL-6 and (**E**) TNF-α in the sera of mice subjected to liver IRI. Sham operated mice were used as control. IRI mice were either treated with saline vehicle or with MIG30 peptide (100 µg 15 min before the reperfusion and again 6 h later, i.v.). CXCL1, CXCL2, CXCL6 and IL-6 were measured at 6 h of reperfusion. TNF-α was measured 12 h post reperfusion. ND = Not detected. The data shown are the mean ± SEM; n is at least 4 mice per group. * *p* < 0.05 vs. sham group; # *p* < 0.05 vs. saline-treated mice.

**Figure 5 ijms-23-09715-f005:**
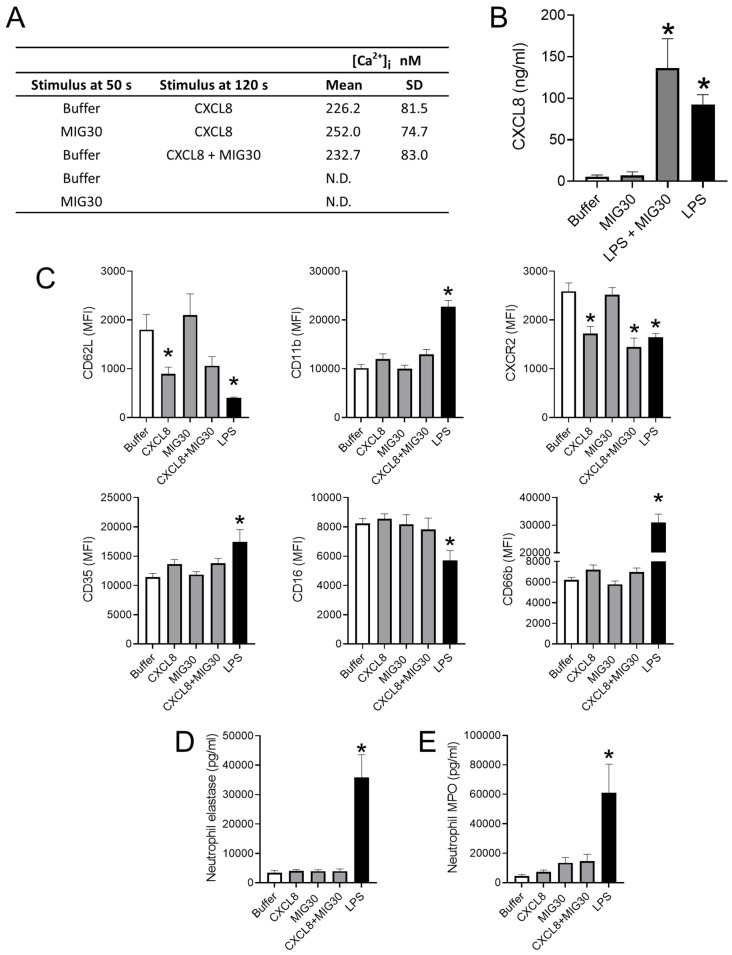
The MIG30 peptide does not affect the responsiveness or activation status of leukocytes towards stimuli. (**A**) Intracellular calcium signal ([Ca^2+^]_i_ in nM) in purified human neutrophils loaded with Fura-2/AM and stimulated with CXCL8 (10 ng/mL) and/or MIG30 (20 µg/mL). Results shown are the mean ± SD increase of the [Ca^2+^]_i_ (nM) of 4 independent experiments. N.D. = Not detected. (**B**) Production of CXCL8 by human PBMCs stimulated with LPS (100 ng/mL), MIG30 (4 µg/mL) or both stimuli combined. Cells were incubated for 24 h with the stimuli, and CXCL8 was measured in the supernatant by ELISA. (**C**) Flow cytometry analysis of purified human neutrophils stimulated for 1 h with LPS (5 µg/mL), CXCL8 (10 ng/mL), MIG30 (20 µg/mL) or a combination of CXCL8 and MIG30. The surface expression of CD62L, CD11b, CXCR2, CD35, CD16 and CD66b was evaluated on neutrophils gated as CD16+CD66b+ cells. Results are presented as mean fluorescence intensity (MFI) ± SEM (*n* = 4). (**D**,**E**) The concentration of (**D**) neutrophil elastase and (**E**) myeloperoxidase (MPO) in the supernatant of purified neutrophils stimulated as in C. Elastase and MPO were quantified by ELISA. Data are shown as mean ± SEM (*n* = 6). * *p* < 0.05 vs. buffer-treated cells.

## Data Availability

Data are contained within the article or Appendix A.

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
