# Peer review of "The GAG-Binding Peptide MIG30 Protects against Liver Ischemia-Reperfusion in Mice"

_ijms, 2022, doi:10.3390/ijms23179715_

Round 1

Reviewer 1 Report

In this article, Caldeira de Oliveira and colleagues explore the therapeutical potential of the GAG-binding peptide MIG30 in liver ischemia-reperfusion (IR) injury. The authors demonstrate that this peptide blocks chemokine binding to GAGs and nicely show that MIG30 protects mice from IR by impairing neutrophil recruitment to the liver.

The paper is very well written and very easy to follow. I do not have any experimental request or suggestion for the authors but I am slightly concerned about the title of the paper and the categorical statement that MIG30 protects from IR by blocking chemokine-GAG interactions. I am not fully convinced that the data included in the manuscript unequivocally demonstrates that this is in fact the therapeutic mechanism, especially because, as the authors show in Fig. 4, MIG30 treatment significantly decreases serum levels of potent neutrophil chemoattractants, which may result in impaired neutrophil recruitment into the liver and protection against IR. I agree with the authors' hypothesis that it is possible that MIG30 binding to endothelial GAGs may lead to accelerated degradation or elimination of chemokines from the blood stream, but this is not tested experimentally in the article. The authors demonstrate that MIG30 does not influence chemokine and cytokine expression by PBMC, but it remains possible that MIG30 may alter cellular responses and chemokine expression by other cell types, like endothelial cells, which are known to present higher levels of surface GAGs than PBMCs. For all this, I suggest the authors to consider changing the title and toning down some of their statements regarding the therapeutic mechanism of MIG30 in their model. 

Author Response

Reply to Reviewer 1

In this article, Caldeira de Oliveira and colleagues explore the therapeutical potential of the GAG-binding peptide MIG30 in liver ischemia-reperfusion (IR) injury. The authors demonstrate that this peptide blocks chemokine binding to GAGs and nicely show that MIG30 protects mice from IR by impairing neutrophil recruitment to the liver.

The paper is very well written and very easy to follow. I do not have any experimental request or suggestion for the authors but I am slightly concerned about the title of the paper and the categorical statement that MIG30 protects from IR by blocking chemokine-GAG interactions. I am not fully convinced that the data included in the manuscript unequivocally demonstrates that this is in fact the therapeutic mechanism, especially because, as the authors show in Fig. 4, MIG30 treatment significantly decreases serum levels of potent neutrophil chemoattractants, which may result in impaired neutrophil recruitment into the liver and protection against IR. I agree with the authors' hypothesis that it is possible that MIG30 binding to endothelial GAGs may lead to accelerated degradation or elimination of chemokines from the blood stream, but this is not tested experimentally in the article. The authors demonstrate that MIG30 does not influence chemokine and cytokine expression by PBMC, but it remains possible that MIG30 may alter cellular responses and chemokine expression by other cell types, like endothelial cells, which are known to present higher levels of surface GAGs than PBMCs. For all this, I suggest the authors to consider changing the title and toning down some of their statements regarding the therapeutic mechanism of MIG30 in their model.

We thank the reviewer for their comments. As suggested, we have changed the title and toned down some statements in the text. The title was changed to:  The GAG-binding peptide MIG30 protects against liver ischemia-reperfusion in mice. Also, the abstract was corrected and the suggestions of “blockage of chemokine-GAG interactions” were removed. The text was toned down in lines 76, 92, 186, 239, 257, 279, 308.

Indeed, the hypothesis that “MIG30 binding to GAGs may lead to accelerated degradation or elimination of chemokines from the bloodstream” was not tested experimentally in the article. Because of that, we replaced the word “suggest” by “speculate” in line 308 and reserved this statement exclusively to the Discussion section.

To further clarify, we have tested in a previous research article if endothelial cells were affected by MIG30 treatment. This was mentioned in the last paragraph of the discussion (Ref 19: Marques, P. E.; Vandendriessche, S.; de Oliveira, T. H. C.; Crijns, H.; Lopes, M. E.; Blanter, M.; Schuermans, S.; Yu, K.; Poosti, F.; Vanheule, V.; Janssens, R.; Boff, D.; Kungl, A. J.; Menezes, G. B.; Teixeira, M. M.; Proost, P., Inhibition of Drug-Induced Liver Injury in Mice Using a Positively Charged Peptide That Binds DNA. Hepatol Commun 2021, 5, (10), 1737-1754.). In the supplementary figure S3 of this reference, we showed that primary human retinal microvascular endothelial cells when stimulated with 1 µM MIG30 produce the same amount of CXCL8 as cells that were incubated with buffer alone. This indicates that MIG30 does not stimulate or inhibit secretion of chemokines at a basal level by these endothelial cells.

Reviewer 2 Report

-The Introduction should be rewrite, to better highlight the involvement of immune system in IRI.

-Figures need to be improved; particularly, Figures 2 and 3 should be ameliorated, increasing resolution and I suggest to insert scale bar in the images.

- The conclusion should be rewrite.

Overall, I suggest the acceptance of the manuscript after these major revisions

Author Response

Response to reviewer 2

The Introduction should be rewrite, to better highlight the involvement of immune system in IRI.

We thank the reviewer for their comments. As suggested, we have added a new paragraph on the involvement of the immune system in IRI in the introduction (line 47). This was added along some reformulation of the introduction text. Moreover, the whole manuscript was checked for typos and mistakes.

-Figures need to be improved; particularly, Figures 2 and 3 should be ameliorated, increasing resolution and I suggest to insert scale bar in the images.

New versions of figures 2 and 3 were added to the manuscript. Moreover, high-resolution PDFs of each figure were uploaded into the IJMS submission platform. The scale bars were already present in the images (always in the lower left corner of the first image in that panel).

- The conclusion should be rewrite.

We added a conclusion statement in line 507.